# New Motherhood Concepts, Implications for Healthcare. A Qualitative Study

**DOI:** 10.3390/ijerph182413118

**Published:** 2021-12-12

**Authors:** Esther Delgado-Pérez, Maria José Yuste-Sánchez, Yolanda Pérez-Martín, Vanesa Abuín-Porras, Isabel Rodríguez-Costa

**Affiliations:** 1Physiotherapy Department, Faculty of Sport Sciences, Universidad Europea de Madrid, Villaviciosa de Odón, 28670 Madrid, Spain; esther.delgado@universidadeuropea.es (E.D.-P.); vanesa.abuin@universidadeuropea.es (V.A.-P.); 2Physiotherapy in Women’s Health (FPSM) Research Group, Physiotherapy Department, Faculty of Medicine and Health Sciences, University of Alcalá, Alcalá de Henares, 28805 Madrid, Spain; marijo.yuste@uah.es; 3Humanization in the Intervention of Physiotherapy for the Integral Attention to the People (HIPATIA) Research Group, Physiotherapy Department, Faculty of Medicine and Health Sciences, University of Alcalá, Alcalá de Henares, 28805 Madrid, Spain; yolanda.perez@uah.es

**Keywords:** motherhood, childcare, society, transition

## Abstract

The aim of this study was to explore the experience of women who take care of their children in postpartum and who desire to be understood by society, with no judgements. For this purpose, a qualitative methodology was followed. In-depth interviews, discussion groups, and an online forum were used for data collection. The participants were Spanish women that had given birth in the past 6 months, and their partners. Healthcare specialists with experience in the topic were also included. Results showed three main categories: lack of priority, self-demand, and self-esteem changes. As a conclusion, the concept of motherhood needs to be redefined, as women feel that they are living under the pressure of being a “perfect mother”. It is important that mothers allow themselves to fail in reaching the imposed requirements. Simple acceptance of motherhood boundaries could help in this transition.

## 1. Introduction

Maternity and motherhood are experienced in different ways by women. Moreover, these experiences are, in many cases, hidden from a society that silences and invisibilizes women [1]. The social forums where women speak freely about their motherhood challenges are scarce. Contextual, economics, social, racial, and ethnic factors determine their experience [2]. Mothers constantly seek to frame themselves in a morally acceptable motherhood [3], which implies them to be devoted to the family and to be successful at work [4].

Motherhood has been idealized, and in many cases is not a fully accepted choice for the mother herself, but a response to a sociocultural pressure that has held women responsible for children care [5]. Data collected from the Spanish National Statistics Institute (INE) showed that women spent 38 h per week in childcare tasks, whereas men spent 23 h [6].

Society has built the ideal concept of “perfect mother” based on a dominant motherhood model, culturally approved, that expects mothers to devote themselves to their children, scarifying their own needs [6,7,8]. Hays (1996) defines this phenomenon as “intensive motherhood”, which would be created around three main concepts: the child as the focus of all attention, breeding and care as the main interest of the mother, and intensive breeding methods [9].

Di Quinzio (1999) labeled the term “essential motherhood”. The mother focuses exclusively and selflessly on childcare [10]. In this model, sensitivity and empathy are inherent features in women, enabling them with ideal abilities for care and sacrifice for others [7]. To some authors, motherhood is experienced as a source of satisfaction with no sacrifice required, inherent for breeding mothers.

In this “intensive motherhood” discussion, the role of parents in the neuropsychological development of their children is pointed out [3]. Most studies focus on the mother–child pair, concluding that this interaction shapes children’s socioemotional development [7,11]. This discussion about children’s development justifies the ideas of the “intensive motherhood” concept. Thus, the contribution of the mother’s role for the improvement of their children’s development and socialization is turned into a core part of their identity [3,12].

Data from a qualitative study carried out in the UK, showed that new mothers were confident that, due to their stimulation, their children would acquire extra in their learning and development process. In this study, participants share their concerns about the fact that “just take care of the baby” was not enough [12]. This dedication forced them to adapt their behaviors to the desired motherhood concept, even during pregnancy, when they adapted their lifestyle and improved their nutrition habits [3,13].

Providing care and being the facilitator of the child’s development have become motherhood-intrinsic requests, having a deep impact on the mother’s experience, especially in new mothers who struggle to adjust their practices in order to satisfy the perceived demands. In many cases, these mothers feel that they are not up to these high self-expectations, which brings on a feeling of guilt [12].

Therefore, mothers are under the pressure of becoming the “perfect mother”, a quite unrealistic expectation, as the requirements of “intensive motherhood” are far too demanding. This has a negative impact on mental wellbeing, self-esteem, and self-efficacy, generating guilty feelings, anxiety, and stress [2,5,14]. Guilt is the most recurrent perceived feeling in mothers, possibly arising from the discrepancy between ideal mother versus real mother. The fear of being judged by others makes this situation more complicated [5,13,14,15].

The aim of this study is to know the experiences, emotions, and feelings related to new motherhood in Spanish women who would like to be understood by society without any judgement.

## 2. Materials and Methods

This is a qualitative phenomenological study with descriptive perspective [16,17] that aims to explore how women experience their postpartum period and the challenges of motherhood. It was carried out between June 2017 and May 2019 in private physical therapy clinics, with a maternal and childcare department in Spain. This study was approved by the Clinical Research Committee of the Hospital de Nuestra Señora del Prado in Talavera de la Reina (CEIm 35/19).

The participants were women in their postpartum period, their partners, and healthcare professionals. Recruitment was carried out through the physical therapy clinics. Inclusion criteria were women who have given birth in the past 6 months and their partners, aged over 18, with physical and psychological ability to understand and participate in the study, and understand and speak Spanish. Type of childbirth and the presence or absence of perineal lesion during labor, previous births, or lactation type were not considered for this study. Exclusion criteria were women and their partners that did not sign informed consent, presence of any systemic disease, neurological problems, and/or cognitive problems, and mother or baby hospitalization/perinatal death in the immediate postpartum. Healthcare professionals were recruited through the snowball method. Professionals with women’s health expertise that were currently attending women in their postpartum period (gynecologists, midwives, and physical therapists) were included. Each participant was asked to sign the informed consent form prior to the study.

The members of the research team were five physical therapists, three specialized in women’s health and two specialized in qualitative research. Data collection was achieved through two interview techniques (in-depth interview and discussion groups) and an observational technique (online forum) in which only the mothers participated. For the development of these, the researcher (EDP) was supported by a semistructured interview guide (Table 1) that was developed after a literature review. These topics were agreed upon by an expert group of four physiotherapists, two of them qualitative research experts and two experts in maternity. The place and the day of the interviews or discussion groups were agreed upon with the participants, being chosen preferably in a quiet room in the most intimate place where the participant feels comfortable, such as the participant’s home and in the case of the discussion groups, in the collaborating physiotherapy centers. Participants only took part in one of the events (interview, focus group, or online forum). The online forum “Sex after Childbirth” was private, and it was active between March 2018 and May 2018. It was linked to a private page of the Facebook interface, and it was opened weekly proposing a debate thread following the guide topics (Table 1) and allowing the participants (which had been invited by mail through a snowball sampling) to interact and feel free to give their opinions under an anonymous profile [18].

Every interview was recorded and fully transcribed. Coding and categorization of the interviews was performed by three members of the research group, individually. For the coding of the fragments, the qualitative analysis software MAXQDA in its 2018 version was used, which facilitated the interactive process [19].

After an in-depth reading, three members of the research team (EDP, IRC, and MJYS) carried out an open, axial, and selective coding [20]. All reports collected (in-depth interview, discussion group, and online forum) were treated equally. Subsequently, they worked together to regroup and agree on the topics through the creation of a code book [21] with the aim of generating a conceptual framework that would explain the phenomenon. Theoretical saturation was achieved with the contribution of DGM31 (discussion group: DG, mother: M, and participant number: 31) in the discussion group carried out with postpartum women. All data were processed following the validation process based on [19] regulated criteria to establish scientific rigor: credibility, transferability, dependence, and confirmability [21,22]. The triangulation of methods, researchers, and locations, the reflection of the research team, the recording of the phenomenon, being as precise as possible, and the description of the methodology developed were guaranteed. Finally, the Standards for Reporting Qualitative Research (SRQR) guidelines [23] and criteria for authors and assessors in the submission and assessment of qualitative research articles [24] were followed to increase scientific rigor.

## 3. Results

Data collection included 28 interviews, 5 focus groups, and one online forum. Thirty-six women and eight partners participated in the study. The mean age of the participant women was 34 years (27–39 rank); the mean age of 8 partners was 38 years (32–48 rank). The mean age of their babies was 4 months and 2 weeks (ranking from 1 month 1 week to 6 months). A total of 50% of participants were new parents, 41% had one more child, and 9% had two more. All the partners were men. Fifteen healthcare professionals participated: six midwives, five physical therapists, and four gynecologists, with an experience mean of 15 years (2–35 rank) (Table 2).

The term “new motherhood” was selected to label the category, and 399 fragments were identified, which were grouped into three main topics: lack of self-priority, self-demand, and self-esteem (Table 3).


Lack of Self-priority


Lack of self-priority describes how women prioritize baby’s, partner’s, or family’s wellbeing, disregarding their own needs. This concept shows how they give more importance to baby, partner, and family wellbeing than to their own needs. It is organized in three subconcepts: baby caring, care for their partners, and lack of self-care.


*“Well, I didn’t manage to get time to make some exercise, I didn’t manage to get time to do anything but being with him (the baby), but neither I wanted to, nor I want to, yet. That is why I am considering whether return to my job or not.” (DGM27, mothers’ discussion group).*


They also prioritize *partner’s care*. In many cases, women describe that, in order to keep their partners satisfied, they yield to sexual demands.


*“[…] is a little more the part in which you think that you have to take care of your partner in that way, because, of course, they don’t experience so many changes in their bodies nor so much hormonal changes” (IM4, in-depth interview with mothers).*


Motherhood demands prevent them from having available time for self-care. When searching for this time, women feel guilty. These emotions are common between the participant mothers, and the health professional themselves are aware of them:


*“I do feel wrong sometimes, when I leave her, like, Damn it! And it is only for an hour, I know it is not a long time….but still…” (DGM25, Mothers’ discussion group).*



*“Of course, nobody tells them so. Everybody tells them to give everything to the baby, right? And they feel they are better mothers if they look untidy while taking care of their baby, so maybe is that precisely what is expected from them, right? Then if, if they tell their own mother or their mother-in-law to take the baby for a little while to have a shower, to sleep for a while, they feel like bad mothers.” (DGPr14, healthcare professionals’ discussion group).*



Self-demand


This topic shows how mothers feel about higher level of demands in order to be up to their expectations. They narrate how they find themselves in a highly demanding stage, as can be found in the following subtopics: “being a supermom”, “frustration”, and “women duties”.

Being a Supermom

Women explain how they feel the duty of *“being a supermom”*. They intend to achieve maximum performance, wishing to be their best, disregarding if they are beyond their limits or their abilities. In the phase they are living, the requirements are higher than in other stages of their lifespan. Feelings such as frustration or resignation arouse, causing further stress.


*“But I have days when is absolutely exhausting. So, when they arrive at 8 pm, I go into the bathroom and cry a little, saying: “I can’t any more, today I can’t take any more…today I am overwhelmed, I am overcome by the situation.” Because you want to cope with everything…and you feel that you are coping with nothing. And that, for me, is the most complicated issue.” (IM1, in-depth interview with mothers).*


These self-demanding feelings are perceived by their partners and the healthcare professionals assisting them.


*“[…] I think that the hardest issue for my wife, if I have to say, more than the hormones, more than any change of these type, was the fact that, having a baby, no matter how good she is, she has not enough time to do everything she wants to do along the day. That is, clearly, more than clearly, the hardest issue for her, above all.” (IP2, in-depth interview with partners).*


Returning to work is a new change to face. Handing over their baby’s care increases negative feelings, as does the self-demand to be equally efficient in their jobs and their new personal lives:


*“Everything has been quite normal, until she returned to work. It is then, when I have seen like a little sensation that everything is overwhelming to her, right? She has returned like, with her issues, her two jobs. Then it has been really, really hard for her. Then, regarding that, I have seen her a little more shaken. To feel that she has not, she cannot do things that she did so easily before, now it is hard for her“. (DGP8, partners’ discussion group).*


Frustration

The ideal of motherhood to which woman are exposed arouses feelings that are shared through their narrations. They especially show their *frustration* about real motherhood.


*“Because people tell you that it is wonderful and they’re super happy, then, why I don’t feel like that. Then you punish yourself, “I want to be ok, and I am not”. But that is because they have told you a reality that is not like that” (DGM36, mothers’ discussion group).*



*“Well… sometimes motherhood is a little…stressing, depending on the expectations that you had before, or not.” (DGPr13, healthcare professionals’ discussion group).*


Women Duty

Self-demanding present in mothers expands itself to their intimate and sexual relationships. Once again, they feel the duty of resuming and maintaining sexual relations with their partners, as a part of their *women duty*.


*“Awkward, I felt awkward. I felt….I don’t mean obliged, because, my poor thing, on the contrary, he has respected me a lot. But it was for me, like a duty that….” (IM11, in-depth interview with mothers).*



Self-esteem


Fragments linked to corporal image, and its impact on their experience as mothers and women were found. Three subtopics were identified: *corporal image, fear of not being likeable, and personal transformation*.

Corporal Image

Some of the changes associated with motherhood focus on the change of *corporal image* perception. Frequently, they show difficulties in assimilating these changes, and on many occasions, it affects their self-esteem.


*“It is like, everything, it is that you no longer recognize yourself, I do not recognize myself, at least […] I cannot look at myself directly in the mirror…” (IM1, in-depth interview with mothers).*


For them, these changes have an impact on their relationships with their partners, on the closeness and sexual relations, which adds to their concerns about their corporal image changes.


*“Well, then….well…wrong. Honestly, wrong. Because, also, him…He expected me to be the same lioness in bed, you know? And…No, I do not any longer.” (IM2, in-depth interview with mothers).*


Fear of Not Being Likeable

The influence of these changes in sexual or simple couple relationships brings feelings of insecurity, arousing the *fear of not being likeable* to their partners as they were before childbirth.


*“And I am, like, afraid of, as I know that he liked me a lot like I was before, I am afraid that he doesn’t like me as I am now, you know? Then, sometimes…” (IM2, in-depth interview with mothers).*


Personal Transformation

Besides negative emotions arising in the participants, some of them showed positive feelings about changes associated to puerperium. These new feelings are generated from an adaptative, coherent perspective towards changes at this stage. It is a personal transformation that would imply the beginning of self-recognition as mothers, women, and partners, empowering their image and self-esteem.


*“This reveals to us unconditional love, it is a unique chance for learning how to love ourselves in an unconditional way, the same as we love the baby. The power of healing our self-esteem and self-love that will surely get us to a good shore in the future.” (FOM13, online forum for mothers)*



*“Well, I think so, I think that it has an impact on her for good […], If this has changed her, it is a good change, for sure.” (DGP8, partners’ discussion group).*


There was a desire to focus the importance of having more real information about postpartum from the healthcare area. Participants stated that this could assist them in normalizing mothers’ experiences and soothe the aroused feelings, avoiding guilt and improving self-care, promoting the concept that, to take care of another being, it is necessary to take care of oneself.


*“And it seems that you even feel guilty and a bad mom about feeling the way you feel or thinking the way you think…I don’t know, I think that maybe it would be necessary to have better information about postpartum and the…the emotional changes that you’re going to suffer or may suffer.” (IM8, in-depth interview with mothers).*


## 4. Discussion

The aim of the present study was to explore how women face motherhood in a high-demanding society. Participants showed their beliefs following the “intensive motherhood” model [9], revealing their lack of self-priority. Scientific evidence has stated that mothers associate full implication in children’s care as one of the main components to become a “good mom” [1,5,7,25,26,27]. Mothers felt that it was their duty and their wish to care and breed, being their main responsibility [25].

Following Swigart’s (1991) point of view, the “good mother” puts her children’s care first, considering motherhood as a source of satisfaction and not requiring any sacrifice [28]. Mothers participating in this study did not express these feelings. Women follow stablished motherhood patterns [5,25], revealing a struggle to match their practices to those currently prescribed [3,12].

Data from several studies reveal a personal discrepancy between the ideal of motherhood and real motherhood, which leads mothers to feelings of frustration and guilt [3,5,7,13]. In the participants’ narrative, a fight to frame themselves into the socially accepted motherhood is observed. This fight makes them feel guilty and ashamed if they have failed to act inside the frame of ideal motherhood [3,5,12,13,15]. Montemurro (2012) stated that women that focus only on taking care of others may be subjected to frustration, stress, and exhaustion [8].

Returning to work is possibly a fact that, again, increases self-demanding in mothers. Re-entry to employment becomes a further stress source. Some studies describe that mothers believe that they are not covering the needs and wishes of their children [14], and sometimes their own wish of keeping their babies close [29]. Aristegui (2018) observed that one of each four participants considered that children suffer more with working mothers [27]. Women in the present study perceived, when returning to work, more demands and lack of efficiency, increasing their emotional distress due to their self-imposed demands.

Another important fact that raises dissatisfaction and stress level is related to the experienced body changes. DeMaria (2019) describes mothers perceiving their own bodies as “strange” [30]. This is possibly due to unreal expectations in women [31]. This conflict with their body image leads to negative emotions and sensations in women that can even be related to conflicts with their partners [4,31,32,33,34]. In the present study, women explained their fear of not being liked by their partners, which causes insecure feelings that are transferred to intimate relationships. Many studies reveal that women after childbirth share the need of feeling desirable in order to improve their self-esteem [8,35,36]. During motherhood transition, sexuality is variable within the first year [37,38]. One of the elements that adds complexity to this topic is the guilt that women feel about not satisfying their partners. Therefore, they yield to their partner’s wish of sexual relations with the added fear of not being likable to them [8,33,39]. Trice-Black stated that some women even fake orgasm in order to flatter their partners [15]. Some authors had observed similar behavior in their study, adapting sexual practices to this new situation [8,40].

Finally, some of the participants described motherhood from a global acceptance and personal transformation perspective. Those are woman that choose to experience motherhood following their own beliefs, and, despite describing motherhood as a complex phase, they could face it. This transformation is also perceived by their partners, who refer to a positive change. Some evidence supports this statement, from changes achieved through challenging experiences [15]; describing the maturity achieved after childbirth [31]; or even acknowledging their bodies’ ability and finding a deeper understanding of their sexuality [8].

It is important to highlight that motherhood models can be rebuilt, becoming a plural concept. Women should choose how to experience their motherhood, ignoring social, religious, or cultural concepts. In order to achieve that, it is essential to deconstruct the “perfect mother” image, supporting and reinforcing the strengths women have in order to empower their self-esteem in the motherhood task [2]. If motherhood was demystified, mothers would probably increase their self-confidence and face the transition to this phase with confidence. This information would allow them to have a holistic approach in healthcare interventions, such as dealing with urogynecology problems in physiotherapy units.

Even though, in this study, validity was controlled through triangulation, it is necessary to assume that the results may be not appliable beyond the study’s sample. Participants were Spanish Caucasic women without economic difficulties. It would have been of interest to spread the sample to adolescents, single mothers, low-income families, or homosexual couples, as it is plausible that their experiences could differ according to their particular characteristics.

## 5. Conclusions

Motherhood implies a natural change in the mother’s priorities, highlighting baby and family care. The motherhood role carries high demands that women and society normalize. Women try to reach an ideal motherhood that raises feelings of frustration. Self-esteem is also affected, influencing physical acceptance and couples’ relationships, bringing further insecure feelings.

Regarding the motherhood process, women who had accepted and experienced it from their own internal reference had undergone a transformation beyond physical or emotional aspects, discovering a new, empowering phase.

Motherhood should be redefined, due to the possibility of women presenting mixed feelings about the experience. It is important that mothers allow themselves to not reach the requirements imposed by society. Acceptance of their limitations would help women during this phase.

## Figures and Tables

**Table 1 ijerph-18-13118-t001:** Interview guide.

**Topis**
How are you experiencing this new stage (after childbirth)?
How do you feel about your body after childbirth?
How do you feel about reinitiating sexual relations?
How are you experiencing the bond with your baby?
How stressful is this new stage?
Could you tell me if you have felt insecure at any point of this new stage?
Do you think there is any factor that influences your satisfaction as a partner?

**Table 2 ijerph-18-13118-t002:** Demographic data.

Demographic Data		n	%
Mothers		36	
Age (years)			
	25–30	2	5.55
	31–35	24	66.66
	36–39	10	27.77
Number of births			
	1	21	58.33
	2	12	33.33
	3	3	8.33
Civil status			
	Married	28	77.77
	Not married	8	22.22
Educational level			
	Secondary Education	6	16.66
	University Education	26	72.22
	MSc or PhD	4	11.11
Partners		8	
Age (years)			
	30–35	2	25
	36–39	3	37.5
	40 or more	3	37.5
Number of children			
	1	4	50
	2	3	37.5
	3	1	12.5
Civil status			
	Married	5	62.5
	Not married	3	37.5
Educational Level			
	Secondary Education	1	12.5
	University Education	3	37.5
	MSc or PhD	4	50
Healthcare Professionals		15	
Midwives		6	40
Physical Therapists		5	33.33
Gynecologists		4	26.66
Age (years)			
	30–35	4	26.66
	35–40	5	33.33
	40 years or more	6	40
Experience (years)			
	Less than 10	6	40
	10–14	3	20
	15–19	2	13.33
	20–24	1	6.66
	More than 25	3	20

**Table 3 ijerph-18-13118-t003:** Category “new motherhood”.

“New motherhood”	Lack of self-priority (86 units of meaning)	Baby care
		Partner care
		Lack of self-care
	Self-demand (155 units of meaning)	“Being a supermom”
		Frustration
		“Being an obligation”
	Self-esteem (158 units of meaning)	Corporal image
		Fear of not being likeable
		Personal transformation

## Data Availability

Data are held securely by the research team and may be available upon reasonable request and with relevant approvals in place.

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
