# Peer review of "New Motherhood Concepts, Implications for Healthcare. A Qualitative Study"

_ijerph, 2021, doi:10.3390/ijerph182413118_

Round 1
Reviewer 1 Report
I thank the authors for the opportunity to review this interesting article, it is an interesting article that collects the challenges faced by women in their motherhood in today's society. However, it would be interesting take into account the following recommendations and respond to the questions posed:
• In the introduction section you talk about a morally accepted motherhood and that it is an idealized event, it would be interesting to give more information, what does this refer to? What is considered a morally accepted motherhood? What is the ideal of motherhood? Be more specific.
• It would be interesting to reformulate the objective of the study, it describes it in a complex way, it should be planted in a concrete, simple and neutral way. It takes assumptions for granted without actually developing the study; On the other hand, it would be interesting to emphasize the role of motherhood within the study.
• Focus on describing aspects of the study currently presenting, introduce elements of the first study that in this case are irrelevant and that generate confusion in relation to the objectives set.
• In the study, he comments that they used in-depth interviews and discussion groups. How many participants were interviewed? What was the basis for making the script of the interview questions? How many discussion groups were held? What questions were raised in the focus groups? Did the interviewed participants also participate in the discussion groups or only a part of them? What criteria did they use to conduct the interviews or discussion group? How long were the interviews? What about discussion groups?
• In the study, you comment that data saturation was reached with participant number 31, why did you continue to expand the sample size until reaching 36 participants?
• How were the results of the study validated? How did you guarantee the reliability of the results?
• What type of analysis did you carry out?
• It would be interesting to develop the first topic in more detail, in a similar way as it does with the rest of the topics, it is brief and should delve into it.
• In the discussion section, the implications in clinical practice of this study, the limitations and future lines of research should be highlighted.
All the best
Author Response
I thank the authors for the opportunity to review this interesting article, it is an interesting article that collects the challenges faced by women in their motherhood in today's society. However, it would be interesting take into account the following recommendations and respond to the questions posed:
- In the introduction section you talk about a morally accepted motherhood and that it is an idealized event, it would be interesting to give more information, what does this refer to? What is considered a morally accepted motherhood? What is the ideal of motherhood? Be more specific.
Authors: Thank you very much for your remark in order to clarify the message; we have included an explanation
- It would be interesting to reformulate the objective of the study, it describes it in a complex way, it should be planted in a concrete, simple and neutral way. It takes assumptions for granted without actually developing the study; On the other hand, it would be interesting to emphasize the role of motherhood within the study.
Authors: Thanks for your appreciations. The aim has been reformulated.
- Focus on describing aspects of the study currently presenting, introduce elements of the first study that in this case are irrelevant and that generate confusion in relation to the objectives set
Authors: Thank you for your valuable comment. This information has been removed.
- In the study, he comments that they used in-depth interviews and discussion groups. How many participants were interviewed? What was the basis for making the script of the interview questions? How many discussion groups were held? What questions were raised in the focus groups? Did the interviewed participants also participate in the discussion groups or only a part of them? What criteria did they use to conduct the interviews or discussion group? How long were the interviews? What about discussion groups?”
Authors: We appreciate your proposed modification. This information has been included in methods section.
- In the study, you comment that data saturation was reached with participant number 31, why did you continue to expand the sample size until reaching 36 participants?
Authors: Thanks for your appreciations. Data saturation was reached with participant 31 analysis that was done after collecting data.
- How were the results of the study validated? How did you guarantee the reliability of the results?
Authors: Thank you for your valuable comment. This information has been highlighted “The triangulation of methods, researchers and locations, the reflection of the research team, the recording of the phenomenon, being as precise as possible, and the description of the methodology developed were guaranteed.”
- What type of analysis did you carry out?
Authors: Thank you for your valuable comment. This information has been highlighted “The triangulation of methods, researchers and locations, the reflection of the research team, the recording of the phenomenon, being as precise as possible, and the description of the methodology developed were guaranteed.”
Authors: Thanks for your appreciations. The order that you propose is much more in line with the current evidence. We have also clarified the concept of "Home programs".
- It would be interesting to develop the first topic in more detail, in a similar way as it does with the rest of the topics, it is brief and should delve into it.
Authors: We appreciate the proposal clarify the information. This paragraph has been modified following your indications.
- In the discussion section, the implications in clinical practice of this study, the limitations and future lines of research should be highlighted
Authors: Thank you for your consideration, we are sorry we were not clear with this information. We add the information requested.
Reviewer 2 Report
The authors conducted a qualitative study to investigate Motherhood in post-partum women. In my opinion "just" a few issues should be "fixed" for publication.
1) Abstract: "Healthcare specialist with experience in the topic were also included". How many specialists were included? The noun is singular and the verb is plural. Moreover, specialists included exactly in what? Be more specific.
2) I would add regarding the opportunity for women of speaking freely, the effect of virtual environments in eliciting high levels of self-disclosure. Among other phenomena, you can refer to the "Stranger on the Internet effect" (https://doi.org/10.1016/j.chb.2015.02.027; https://doi.org/10.3390/fi13050110).
3) I would also recommend improving the paper by referring to the guidelines for assessing qualitative research developed by Kitto et al. (2008). Authors should justify their work better. Why is a qualitative approach the best option to answer this specific question? Moreover, why your type of analysis was chosen and no other techniques/designs?
Author Response
The authors conducted a qualitative study to investigate Motherhood in post-partum women. In my opinion "just" a few issues should be "fixed" for publication.
1. Abstract: "Healthcare specialist with experience in the topic were also included". How many specialists were included? The noun is singular and the verb is plural. Moreover, specialists included exactly in what? Be more specific
Authors: We greatly appreciate your initial comments. They greatly encourage us to continue working in this line. The information has been explained in Table 2.
2) I would add regarding the opportunity for women of speaking freely, the effect of virtual environments in eliciting high levels of self-disclosure. Among other phenomena, you can refer to the "Stranger on the Internet effect" (https://doi.org/10.1016/j.chb.2015.02.027; https://doi.org/10.3390/fi13050110).
Authors: We appreciate your proposed modification. This reference has been included in methods section.
3. I would also recommend improving the paper by referring to the guidelines for assessing qualitative research developed by Kitto et al. (2008). Authors should justify their work better. Why is a qualitative approach the best option to answer this specific question? Moreover, why your type of analysis was chosen and no other techniques/designs?
Authors: Thanks for the consideration. A qualitative research was developed because research think it is the best way to know participant´s experiences about motherhood. The reference has been included in methods section.
Round 2
Reviewer 1 Report
Dear authors
Thanks for giving me the opportunity to review again, now I think it could be published. In the future, in order to facilitate the revision in future articles, also add in the response letter the changes that you include in the manuscript.
All the best